# Astaxantin and Isoflavones Inhibit Benign Prostatic Hyperplasia in Rats by Reducing Oxidative Stress and Normalizing Ca/Mg Balance

**DOI:** 10.3390/plants10122735

**Published:** 2021-12-12

**Authors:** Alexander L. Semenov, Ekaterina A. Gubareva, Elena D. Ermakova, Anastasia A. Dorofeeva, Irina A. Tumanyan, Ekaterina A. Radetskaya, Maria N. Yurova, Saied A. Aboushanab, Osman N. Kanwugu, Elena I. Fedoros, Andrey V. Panchenko

**Affiliations:** 1N.N. Petrov National Medical Research Center of Oncology, Leningradskaya str, 68, 197758 St. Petersburg, Russia; gubareva1984@gmail.com (E.A.G.); helenermakova@mail.ru (E.D.E.); donastya94@mail.ru (A.A.D.); itumanyan@mail.ru (I.A.T.); radison-kat@mail.ru (E.A.R.); yumarni@gmail.com (M.N.Y.); elenafedoros@gmail.com (E.I.F.); ando.pan@gmail.com (A.V.P.); 2Institute of Biomedical Systems and Biotechnology, Peter the Great St. Petersburg Polytechnic University, Polytechnicheskaya, 29, 195251 St. Petersburg, Russia; 3SCAMT Institute, ITMO University, Lomonosova St. 9, 191002 St. Petersburg, Russia; 4Institute of Chemical Technology, Ural Federal University Named after The First President of Russia B. N. Yeltsin, Mira 19, 620002 Yekaterinburg, Russia; sabushanab@urfu.ru (S.A.A.); okanvugu@urfu.ru (O.N.K.)

**Keywords:** benign prostatic hyperplasia, testosterone, isoflavones, astaxanthin, magnesium, calcium, oxidative stress, rats, kudzu, *P. rhodozyma*

## Abstract

Benign prostatic hyperplasia (BPH) is a common pathology among aging men. Despite the broad pharmacological interventions, the available remedies to treat BPH are yet not devoid of side effects. Herbal compounds are suggested to be an alternative option for the BPH treatment. In our study, we evaluated the effect of kudzu isoflavones and astaxanthin on the BPH animal model. The animals were randomly divided into five groups: control; testosterone-induced BPH group; and three BPH-induced groups, which received intragastrically for 28 days finasteride (5 mg/kg) as a positive control, isoflavones (200 mg/kg), and astaxanthin (25 mg/kg). BPH was induced by castration of animals and subsequent subcutaneous injections of prolonged testosterone (25 mg/kg). Prostate index and histology, biochemical parameters, and antioxidant activity were evaluated. A significant decrease in prostate weight, immunohistochemical markers, and normalization of prostate Ca/Mg ratio was found in all treatment groups. Astaxanthin treatment also resulted in decreased epithelial proliferation and normalized superoxide dismutase activity. In conclusion, both isoflavones and astaxanthin inhibited BPH development at a level comparable to finasteride in terms of prostate weight, prostatic epithelium proliferation, and prostate tissue cumulative histology score. These results suggest that isoflavones and especially astaxanthin could serve as a potential alternative therapy to treat BHP.

## 1. Introduction

Benign prostatic hyperplasia (BPH) is considered the most common pathological manifestation among aging men [1]. BPH is associated with worse quality of life due to various lower urinary tract symptoms and there is data pointing at the possibility of increased risk of prostate cancer [2,3]. This disease has high personal and healthcare costs, both direct medical and indirect losses in daily functioning [4]. Although the molecular biological mechanisms influencing the etiology of BPH have not been yet elucidated, prostate development and growth are largely dependent on testosterone and 5α-dihydrotestosterone (DHT), the more active form of testosterone. Both hormones bind to a specific androgen receptor to form a complex that can regulate gene expression [5]. The activity of 5α-reductase is also implicated in benign prostatic hyperplasia [6]. In addition, oxidative stress [7,8] and chronic inflammation [9] is believed to be associated with prostate cell hyperproliferation and histopathology. Currently, alpha-blockers and synthetic inhibitors of 5α-reductase, finasteride, and dutasteride, are registered for BPH management in clinical practice [10,11,12], but these drugs have a number of known adverse effects.

A large group of natural compounds is commonly used for BPH prevention and controlling the lower urinary tract symptoms associated with BPH. For instance, botanicals (*Serenoa repens*, *Pygeum africanum*, *Urtica dioica*, *Cucurbita pepo*, *Epilobium spp*., *Lycopersicum esculentum*, *Secale cereale*, *Roystonea regia*, *Vaccinium macrocarpon*) and nutraceuticals (isoflavones, astaxanthin, lycopene, selenium, and β-Sitosterol) were reported for low toxicity and pronounced efficacy against BPH [10,13,14], and also there is data for beneficial action of Kaempferol, Myricetin, and Fisetin on prostate and bladder cancer [15] Efficacy of different plant extracts was demonstrated in experimental animal models of BPH [16,17]. Phytoestrogens are plant constituents that were documented for their broad therapeutic interventions, e.g., anticancer, anti-inflammatory, and immunomodulatory effects due to their isoflavones rich content [18]. These phytoestrogens have substantially confirmed their ability to influence testosterone levels and may possibly exert a protective therapeutic effect against the development of prostate cancer [19,20].

Among the wide range of medicinal plants, the Leguminosae family is a rich source of isoflavones [21]. Soybean is an important food crop, which predominantly contains high quantities of isoflavones [22]. Most common isoflavones include daidzein, genistein, biochanin A, and formononetin. These bioactive compounds can act as agonists or antagonists to human hormones [23] and have variable effects on growth factor inhibition [24], antioxidant properties [25], and 5α-reductase reductase activity [26]. Importantly, isoflavones are often found as glycosides and are converted to aglycones by the action of gut microflora [21].

*Pueraria lobata* or kudzu is a medicinal plant that contains a number of phytoestrogens, particularly isoflavones [27]. Daidzein, genistein, puerarin, formononetin, and biochanin are considered the most abundant active metabolites that could be extracted from *Pueraria lobata* [28,29]. These active ingredients may possibly exhibit potential effects against BPH induced by testosterone in in vivo studies.

Other dietary components with positive effects in patients with BPH and prostate cancer with extensive investigation are carotenoids, such as lycopene and astaxanthin [14,30,31,32]. Astaxanthin is a red-orange liposoluble pigment belonging to a subset of carotenoids known as the xanthophylls. It is produced primarily by some algae, yeasts, and bacteria, notably, *Haematococcus pluvialis*, *Phaffia rhodozyma* (also known as *Xanthophyllomyces dendrorhous*), and *Paracoccus spp*. respectively [33,34]. Astaxanthin production through microalgal *Haematococus pluvialis* is well studied [35] and is presently industrially produced in large scale [36]. Astaxanthin is renowned for its potent and superior antioxidant activity among natural bioactive substances. It is also acclaimed for its numerous biological activities including anticancer, anti-inflammatory, and immuno-modulative activities [34,37]. Of interest to this present study, astaxanthin has been shown to exhibit antiproliferative and proapoptotic effects in some models [38,39]. Moreover, Anderson [40] previously reported that astaxanthin could inhibit 5α-reductase activity and growth of LNCap-FGC cells. The effect of astaxanthin on BPH in in vivo model was not previously assessed according to the best of our knowledge.

The aim of current study was to evaluate the in vivo efficacy of isoflavones of kudzu (NADES extract of *Pueraria lobata* roots), containing daidzein, genistein, puerarin, formononetin and biochanin A, and astaxanthin (*Phaffia rhodozyma* extract) on testosterone-induced BPH in rats. Prostate weight/index and histology/immunohistochemistry, biochemical parameters of serum and prostate tissue, and antioxidant activity served as endpoints.

## 2. Results

### 2.1. Assessment of Body Weight and Prostate, Liver, and Heart Index

Surgical castration followed by the administration of testosterone increased prostate weight by about 2.8 times compared with intact animals (180 ± 18 vs. 508 ± 34 mg/100 g body weight *p* < 0.001, Table 1). The greatest increase in 3.2 times was observed in the ventral lobes (66 ± 10 vs. 212 ± 18 mg/100 g body weight, *p* < 0.001).

In the positive control group (BPH+finasteride), the weight of the prostate gland of the rats was 25% lower than in the BPH group (382 ± 14 mg/100 g, *p* < 0.001). Both tested substances showed effects comparable to finasteride in terms of reducing the weight of the prostate—459 ± 34 mg/100 g for isoflavones (*p* < 0.05 vs. BPH) and 443 ± 14 mg/100 g for astaxanthin (*p* < 0.01 vs. BPH). The most significant influence of the drugs was observed also for ventral lobes—finasteride decreased their weight down to 148 ± 8 mg/100 g compared to 212 ± 18 mg/100g in BPH group (*p* < 0.001); isoflavones—down to 166 ± 19 mg/100 g (*p* < 0.01 vs. BPH); astaxanthin—down to 178 ± 12 mg/100 g (*p* < 0.01 vs. BPH). Despite significant changes in the body weight of animals, a study of the organ-weight coefficients of the heart and liver showed no significant toxic effects on these organs. Nevertheless, in all groups with BPH, there was a non-significant tendency for the average liver mass to be less than in the control, and the heart mass to be greater.

### 2.2. Histopathological Examination and Assessment of Proliferation Markers (H3Ser10)

The results of histological examination of prostate ventral lobes are shown in Figure 1. The development of prostatic hyperplasia was evident by the significant increase in the area of the epithelium of the ventral lobes of the prostate gland. The administration of the comparison drug finasteride and the astaxanthin significantly reduced this indicator compared with the BPH control group. Evaluation of the proliferative activity of histone H3 phosphorylated on Ser10 showed that BPH is largely associated with an increase in the prostatic epithelium proliferative activity. The administration of all investigated substances, including finasteride, inhibited the proliferative activity of prostatic epithelium compared to the BPH group. However, in all experimental groups, proliferative activity of prostatic epithelium was found higher than in the intact control group. Pathomorphological examination of the ventral lobes showed a significant decrease in the value of the cumulative score of prostate damage [41] in all experimental groups of animals, including those treated with finasteride.

### 2.3. Assessment of Biochemical Analysis

Biochemical analysis of prostate tissue revealed metabolic changes of Ca and Mg (Table 2). While BPH induction caused non-significant counter-directional shifts in their content (increase from 49.0 ± 16.5 to 62.6 ± 15.9 for Ca, *p* > 0.05 vs. control; decrease from 62.2 ± 11.8 to 53.1 ± 15.7 for Mg, *p* > 0.05 vs. control), the ratio of these macronutrients changed significantly from 0.8 ± 0.2 to 1.3 ± 0.3, *p* < 0.001 vs. control. The administration of all investigated substances normalized the balance of these elements—to 1.0 ± 0.1 for finasteride and to 0.8 ± 0.1 for both isoflavones and astaxanthin (*p* < 0.05 vs. BPH for all groups).

### 2.4. Assessment of Blood Count, Antioxidant Activity, Biochemistry, and ELISA Assay

Induction of BPH was accompanied by significant blood count changes in experimental animals (Table 3). Red blood cells count and hematocrit increased in all groups induced with BPH which is consistent with the effects of short-term course administration of testosterone [42]. Moreover, white blood cell counts significantly decreased at the end of the experiment (9.9 ± 0.9 × 10^9^/L) in the BPH group compared to the control group (16.2 ± 1.9 × 10^9^/L) mainly due to the lymphocytic fraction (5.9 ± 0.6 × 10^9^/L in BPH group and 12.6 ± 1.5 × 10^9^/L in control group, *p* ˂ 0.05). It was shown that androgens exert inhibitory effects on B-lymphopoiesis which may be mediated through androgen receptors expressed in bone marrow stromal cells [43]. Treatment with comparator drug finasteride did not affect the above-mentioned indicators while the exposure to herbal preparations contributed to the normalization of lymphocytes count and significantly increased granulocytes count compared with the control animals. When using isoflavones of kudzu root, the level of leukocytes and lymphocytes in the peripheral blood of experimental animals significantly differed from those in the BPH group and approached the values of the control group (WBC 14.2 ± 1.1 × 10^9^/L, Lymph 9.4 ± 0.8 × 10^9^/L).

Biochemical analysis of blood serum (Table 4) revealed that the glucose level decreased (9.2 ± 1.4 mmol/L vs. 7.2 ± 1.4 mmol/L, *p* < 0.01) in comparison with the intact control in the BPH group. Calcium content (2.8 ± 0.3 mmol/L vs. 3.4 ± 0.6 mmol/L, *p* < 0.05) and magnesium content (0.9 ± 0.1mmol/L vs. 1.5 ± 0.8 mmol/L, *p* < 0.05) increased in comparison with the intact control in the BPH group. Nevertheless, the ratio of these parameters did not change. In addition, BPH was accompanied by a significant increase in serum testosterone levels (13.5 ± 4.2 ng/mL vs. 3.4 ± 1.9 ng/mL *p* < 0.001 vs. intact control). Finasteride, isoflavones, and astaxanthin did not affect glucose levels compared to the BPH group but normalized the calcium and magnesium levels in all three groups. Isoflavones did not affect testosterone level. However, finasteride significantly increased the level of testosterone (18.2 ± 3.0 ng/mL *p* < 0.05 vs. BPH group) and the same effect was observed for astaxanthin (17.9 ± 3.3 ng/mL *p* < 0.05 vs. BPH group).

Decreased relative SOD activity in blood serum (Table 4) indicated the development of oxidative stress in animals with BPH (67 ± 7% vs. 100 ± 8% in the control group, *p* < 0.05). Both in the BPH + finasteride (98 ± 6%) and BPH+astaxanthin (111 ± 13%) groups, there was complete normalization of the SOD activity (*p* < 0.05 vs. BPH group). In animals treated with isoflavones, the average SOD activity level was recorded in the range between the value of the control and BPH group (*p* > 0.05).

## 3. Discussion

The development of BPH was accompanied by the increase in prostate gland weight, epithelial proliferation, oxidative stress, and blood serum testosterone level. Finasteride at a dose of 5 mg/kg produced a significant inhibitory effect on the development of BPH which was evident in the decrease of prostate weight, epithelial proliferation, and the increase in blood serum testosterone and decreased oxidative stress marked as SOD relative activity. The increase in serum testosterone is consistent with another study where finasteride (1 mg/kg) and isoflavones were administered to rats in a BPH model [29]. However, in other findings where finasteride (10 mg/kg) was administered to Wistar rats induced with BPH, there was a decrease in serum testosterone and DHT compared with the BHP group [44]. Both prostatic and serum levels of DHT were shown to be decreased in rats treated with finasteride at 0.8 mg/kg/day [45] in a BPH animals model. However, in our study, we did not observe changes in the testosterone level in prostate of rats treated with finasteride. We indicated higher testosterone levels in blood serum following finasteride and astaxanthin treatment, which could be explained by the inhibition of the 5a-reductase enzyme and less conversion of testosterone to DHT in prostatic epithelial cells, but the level of DHT was not assessed in our study. Serum testosterone elevations are known to occur with the administration of finasteride and dutasteride but the values typically remain within the normal laboratory range [46].

In the present study, both isoflavones and astaxanthin inhibited BPH development in rats with comparable efficacy with finasteride in terms of prostate weight, prostatic epithelium proliferation and prostate tissue cumulative histology score. However, only astaxanthin yielded a significant effect on epithelial area and testosterone level in blood compared to finasteride.

Ca/Mg disbalance was shown to play a role in BPH development in humans [47]. Increasing intranuclear Ca has been observed with advancing age and is believed to have pathologic significance in prostate growth disorders [48]. The increase in Ca/Mg ratio was found to be associated with prostatic epithelium proliferation. However, among other metals, the concentrations of Ca and Mg are significantly higher in the malignant prostate compared to the benign [49] which could be essential for the initiation/progression of prostate cancer [50]. On the other side, the decrease in Ca/Mg ratio is mainly associated with apoptosis [51]. There were no differences between finasteride, isoflavones, or astaxanthin effect on calcium–magnesium balance. The administrated three drugs decreased Ca and Mg levels in blood serum and normalized Ca/Mg ratio in prostate tissue along with the decreased proliferative rate of prostatic epithelial cells marked by H3Ser10 expression.

BPH development was associated with oxidative stress as assessed by SOD activity. Astaxanthin was observed to be superior to kudzu isoflavones at restoring SOD activity and probably ameliorating oxidative stress. This is consistent with the astaxanthin inhibition of intracellular reactive oxygen species production during autophagic cell death induced by the estrogenic endocrine-disrupting chemical bisphenol A in normal human dermal fibroblasts [52]. Carotenoid astaxanthin was also reported to inhibit 5α-reductase and decrease the growth of human prostatic cancer cells in vitro [40].

The effect of isoflavones to ameliorate BPH may be due to their ability to inhibit 5α-reductase. Isoflavones and O-methylated isoflavones inhibited rat prostate testosterone 5α-reductase, where genistein, biochanin A, equol, and 3′,4′,7-trihydroxyisoflavone had considerably higher inhibitory effects whereas daidzein, formononetin, glycitein, prunetin, ipriflavone, and 4′,7-dimethoxyisoflavone had lower inhibitory effects [53]. Extract of *Pueraria mirifica* of the family *Leguminosae* containing daidzein, diadzin, genistin, genistein, and puerarin was revealed to reduce benign prostate hyperplasia in Sprague Dawley rats. The effect was comparable to daidzein or genistein single administration [29]. Similarly, the inhibition of 5α-reductase after administration of kudzu root in combination with cinnamon has been investigated in mice with prostatic hyperplasia [54]. Thus, the effect of kudzu root isoflavones may be related to the activity of their genistein, biochanin A, daidzein, and formononetin.

In humans, isoflavones consumed with diet have shown health benefit effects [55,56,57]. Dietary intake of isoflavone showed a strong inverse association with the risk of lower urinary tract symptoms in a large prospective cohort study of 2000 Chinese elderly men. Subjects with dietary total isoflavone of more than 5.1 mg were significantly less likely to suffer from more severe lower urinary tract symptoms [58].

The efficacy and safety of soy isoflavones in controlling the symptoms and signs of lower urinary tract symptoms due to BPH were assessed in a randomized clinical trial with 176 participants. Results showed only slight superiority of isoflavones over placebo over 12 months with otherwise surprising beneficial effects in both groups, as well as, efficient tolerability of isoflavones [59]. Meta-analysis of randomized controlled trials of soy and soy isoflavones in prostate cancer supported epidemiological findings of their potential role in prostate cancer risk reduction with a good safety profile shown for isoflavones supplementation [60].

## 4. Materials and Methods

### 4.1. Materials

#### 4.1.1. NADES (Natural Deep Eutectic Solvent) Extract of Pueraria Lobata Roots

Dried kudzu roots (Pueraria lobata) were purchased from Xi’an Sgonek Biological Technology (Shaanxi Sheng, Xi’an, China). The roots were pulverized to a homogeneous fine powder and powder was extracted using NADES and ultrasound-assisted extraction (UAE) technology following the method described by Duru et al. (2020) with little modifications [61,62]. Briefly, extraction of isoflavones from dried kudzu root was carried out using NADES, comprising choline chloride and citric acid (1:2, mol/mol), at a solvent volume (mL) to the material ratio (g) of 20:1. The final mixture containing 20% of water was used for extraction using ultrasonic power 580 W at frequency 37 kHz and temperature 60 °C for 180 min. The suspension was then centrifuged at 6000 rpm and the liquid extract was fractionated using ethyl acetate and concentrated with a rotatory evaporator until complete dryness. The final extract was then diluted with methanol and the concentrations of isoflavones in the final solution were determined by high potential liquid chromatography (HPLC) using calibration curves.

Isoflavone content was analyzed using an Agilent Poroshell 120 EC-C18 (3.0 × 100 mm × 2.7 µm) reverse stationary phase column with an additional 5 mm guard column on the Agilent 1260 Infinity II system liquid chromatography with UV/Vis detector (Agilent, Santa Clara, CA, USA). The binary mobile phase consisted of 0.1 % (*v*/*v*) acetic acid in water (solvent A) and 0.1 % acetic acid (*v*/*v*) in methanol (solvent B) was used. Linear gradient elution of solvent B was applied starting from 5% up to 100% over 20 min, then kept at 100% over 1,5 min and returned to initial over 3,5 min with a flow rate of 0.7 mL/min. The temperature of the column was kept at 30 °C. Isoflavones were separated on a reversed stationary phase and detected with UV at 254 nm. The identification of the chromatographic peaks of isoflavones extracts was carried out based on the retention times and UV spectra of the peaks corresponding to the reference standards of daidzein, genistein, puerarin, formononetin, and biochanin. The concentration of the total isoflavones in NADES extract of kudzu roots after ethyl acetate fractionation was 30%, where puerarin (27.3%), daidzein, (2.5%,) genistein (0.04%), and formononetin (0.17%). Principal isoflavone compounds in extracts used for the study are shown in Table 5.

#### 4.1.2. Extract of *Phaffia rhodozyma*

Biomass of *P. rhodozyma* was harvested from several growth experiments. Extraction of astaxanthin was carried out as described by Kanwugu et al. [33] with some modification. Frozen biomass was thawed at room temperature for about 1 h and suspended in DMSO at a ratio of approx. 1:4. The suspension was subjected to ultrasonication (at 80 kHz and 100% power; Elmasonic P 60 H (Elma Schmidbauer GmbH, Singen, Germany)) in a water bath (set at 35 °C) for 30 min. About 20 mL of the suspension was apportioned into 50 mL centrifuge tubes and an equal volume of petroleum ether (40–60 fraction) was added and astaxanthin extracted by rigorous shaking for 10 min. The pigment-containing petroleum ether phase was separated by centrifuging at 4000× *g* for 10 min and pooled together. The extraction was repeated by adding fresh petroleum ether to the remainder of the content of the centrifuge tube and the extraction process was repeated until a colorless petroleum ether phase was obtained. The astaxanthin-containing petroleum ether phase was then concentrated to about 150 mL on a rotary evaporator at 35 °C. The concentrated extract was washed thrice with distilled water, each time 100 mL of 20% NaCl solution was as well added to aid phase separation. Anhydrous NaSO_4_ was added to the final petroleum ether phase and filtered. The petroleum ether was then completely evaporated, and the extract was transferred into air-tight containers wrapped with aluminum foil and stored at −18 °C for further use.

An aliquot of the extract after evaporation was redissolved in HPLC grade acetonitrile and the astaxanthin content was quantified using HPLC (Shimadzu LC-20AD HPLC device equipped with a Shimadzu SPD-20AUV/Vis detector (Shimadzu, Kyoto, Japan)) as described by Kanwugu et al. [33]. The study extract obtained from *P. rhodozyma* consists of 76.6 % (*W*/*V*) astaxanthin.

### 4.2. Animals

Sixty male Wistar rats purchased from “Rappolovo” animal facility, Leningrad Region, Russia were used in the experiment. The weight of animals at the commencement of the study was determined (279 ± 23 g). Animals were kept in a conventional polycarbonate cage 1291H (Tecniplast, Buguggiate, Italy) in a 12 h light/dark cycle at 21–23 °C with 20–55% average humidity and with ad libitum access to pelleted laboratory chow (Laborotorkorm Ltd., Moscow, Russia) and tap water.

### 4.3. Experimental Design

Rats were randomly subdivided into five groups. Control animals (*n* = 12) were intact and passed necessary placebo administrations. In the other four groups, benign prostatic hyperplasia was induced as described by Bespalov et al. [63]. Briefly, rats underwent surgical castration under inhalation anesthesia (day 1). Starting from day 7, animals were subcutaneously injected with prolonged testosterone (Omnadren 250^®^, Jelfa, Jelenia Góra, Poland, 25 mg/kg, every second day, 7 times totally) and administration of tested substances began. All tested substances were dissolved in refined sunflower oil to administer 0.5 mL solution per rat by gavage. The course of treatment with test substances constituted 28 days. Animals of the second group (BPH, *n* = 12) were treated with a placebo (daily gavage of refined sunflower oil). The third group (BPH + finasteride, *n* = 12) served as a positive control and was treated by finasteride (OBL Pharm, Obolensk, Russia, 5 mg/kg, daily gavage). The fourth group (BPH + isoflavones, *n* = 12) was treated with isoflavones of kudzu roots (200 mg/kg, daily gavage). The fifth group (BPH + astaxanthin, *n* = 12) was treated with astaxanthin (25 mg/kg, daily gavage). The study was terminated by the 36th day and animals were euthanized by the CO_2_ inhalation method. Blood samples for biochemistry analysis were collected by terminal cardiac puncture and the targeted organs were subjected to histopathological examination.

### 4.4. Measurement of Prostate, Liver, and Heart Index

Animals were weighed and an autopsy was performed with excision of the organs, visually free from adhering tissues. The anterior lobes of the prostate were separated from the seminal vesicles, the dorsolateral and ventral lobes of the prostate were cleared of adipose tissue, and the bladder was removed. The dorsolateral part of the prostate, 2 ventral and 2 anterior lobes of the prostate were weighed separately on an analytical balance (A&D HR-150AZG, Tokyo, Japan). Additionally, the liver and heart were excised and weighed. Organ index was calculated in accordance with the following formula: prostate index = prostate weight (mg)/body weight (g). Organ weight (mg) to body weight (g) ratio was multiplied by 100 and the mean absolute organ to body weight ratios were calculated and compared.

### 4.5. Histopathological Evaluation and Immunohistochemical Detection of H3Ser10

Ventral prostate samples were fixed in 10% neutral buffered formalin. The tissues were processed according to the standard techniques, embedded in paraffin, and then sectioned. Staining was performed with hematoxylin-eosin. Analysis was done using Nikon Eclipse Ni-U microscope (Nikon Corporation, Tokyo, Japan) with a digital camera and NIS-Elements Br software (version 4.30.00; Nikon Corporation). The evaluation of prostatic hyperplasia was performed according to the protocol proposed by Scolnik et al. [41], with minor modifications (Appendix A). In the hematoxylin-eosin-stained ventral prostate lobes sections, the shape of the acini and lumen, the stroma, the type and organization of epithelial cells, proliferation, etc., were assessed on an ordinal scale. Epithelial area was measured in whole ventral lobes as described by de Amorim Ribeiro et al. [64] using ImageJ software (NIH, Bethesda, MA, USA).

Ventral prostate tissues sections were deparaffinized in xylene and immunohistochemically stained with proliferation maker H3Ser10 histone antibodies (sc-8656R, Santa Cruz Biotechnology, Dallas, TX, USA) and the percentage of positively stained nuclei was counted (ImageJ software).

### 4.6. Blood Count

Blood counts were performed in samples collected at the end of study. Blood samples (20–40 μL) were taken from the lateral tarsal vein and collected in test tubes (MiniCollect^®^; Greiner Bio-One International GmbH, Kremsmünster, Austria) containing K3EDTA. Blood analysis was performed on a Mindray BC-2800Vet Hematology Analyzer (Shenzhen Mindray Bio-Medical Electronics Co., Ltd., Shenzhen, China).

### 4.7. Prostate Tissue Probe Preparation

Prostate tissue (ventral lobes) was taken during autopsy and immediately frozen in liquid nitrogen. Samples were mechanically homogenized using liquid nitrogen and diluted in 1.0 mL TBST buffer (tris-buffered saline and Polysorbate 20). The homogenates were centrifuged at 12,000× *g* for 20 min at 4 °C and supernatants were used for further analysis.

### 4.8. Biochemical Analysis, Antioxidant Activity Evaluation, and Testosterone Assay

Biochemical analysis of serum and prostate tissue supernatants was performed on a Konelab 20 analyzer (Thermo Scientific, Vantaa, Finland) using kits for the measurement of total protein, cholesterol, triglycerides, glucose, calcium, magnesium (SPC “Eco-service”, St. Petersburg, Russia) according to the manufacturer’s protocols. The activity of antioxidant enzyme superoxide dismutase (SOD) in red blood cell lysate was evaluated on a Konelab 20 analyzer (Thermo Scientific, Vantaa, Finland) according to the protocol described earlier [65] and expressed as relative activity from the control group which is taken as 100%.

Enzyme-linked immunosorbent assay (ELISA) was performed using a testosterone ELISA kit following the manufacturer’s protocols (DRG, Marburg, Germany). The optical density was measured at 450 nm using iMark microplate absorbance reader (BioRad, Hercules, CA, USA).

### 4.9. Statistical Analysis

Statistical analysis was performed using GraphPad Prism 8 software (GraphPad Software, San Diego, CA, USA). Anderson–Darling test was used to assess the normality of the data. The significance of differences between groups was tested using one-way Welch ANOVA test with post-hoc multiple comparisons corrected by controlling false discovery rate (FDR; two-stage linear step-up method of Benjamini, Krieger, and Yekuteili and Q set as 0.05). Data presented as mean with SEM.

### 4.10. Study Limitations

The current study may have certain limitations. The BPH induction scheme differs from the mainstream models (higher intermittent dose of prolonged testosterone vs. lower dose daily testosterone injections). It might have had effect on severity of induced hyperplasia, but not on relative efficacy of studied substances compared to reference drug. Instead of herbarium deposition that is considered the best practice for such studies, we used HPLC fingerprint to ensure consistency of study samples within and among the studies.

## 5. Conclusions

The administration of isoflavones and astaxanthin, and finasteride, restored the balance of biochemical parameters and histopathological images of prostatic tissues. Both isoflavones and astaxanthin inhibited BPH development in rats with comparable efficacy to finasteride in terms of prostate weight, prostatic epithelium proliferation, and prostate tissue cumulative histology score. However, only astaxanthin yielded a significant effect on epithelial area and testosterone level in blood compared to finasteride. Inspired by the findings which demonstrated its superior efficacy over kudzu roots, astaxanthin is recommended to be a promising agent for the prevention and treatment of BPH. It was concluded that isoflavones and astaxanthin supplementation could inhibit testosterone-induced BPH development in rats which may be linked to their ability to inhibit 5α-reductase. These findings may prompt the development of isoflavones and astaxanthin extracts that could be a significant step towards the prevention of oxidative stress and further complications of BPH.

## Figures and Tables

**Figure 1 plants-10-02735-f001:**
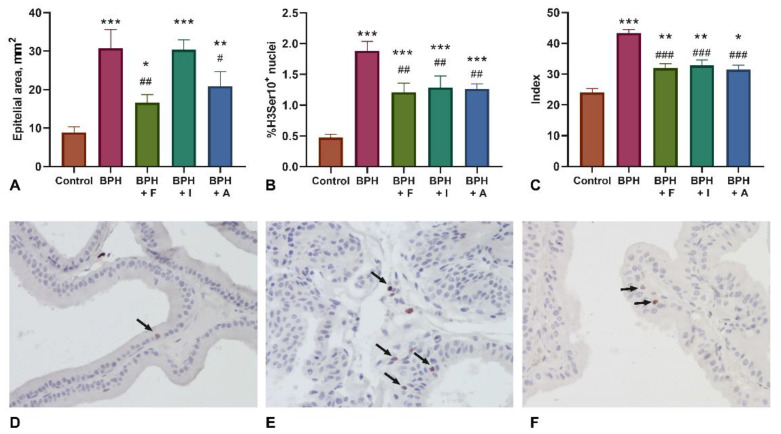
Histopathological assessment in ventral lobes of prostate. (**A**) Area of prostate ventral lobe epithelium. (**B**) Percent of histone H3Ser10 positive nuclei of epithelial cells. (**C**) Cumulative score (units) of BPH histopathology [41]. (**D**–**F**) H3Ser10 immunohistochemical staining in control, BPH, and BPH+A groups, respectively, arrows indicate positive nuclei staining. *, **, ***—*p* < 0.05, *p* < 0.01, *p* < 0.001 compared to control group (placebo), #, ##, ###—*p* < 0.05, *p* < 0.01, *p* < 0.001 compared to BPH group; assessed using ANOVA. Abbreviation: BPH—benign prostatic hyperplasia; F—finasteride; I—kudzu isoflavones; A—astaxanthin.

**Table 1 plants-10-02735-t001:** Body weight and organ/body weight ratio of rats with BPH and treated with finasteride, isoflavones, or astaxanthin at the study end.

Group	Mean Animal Weight (g)		Organ/Body Weight Ratio at the End of the Study (mg/100 g)					
	at the Start of the Study	at the End of the Study	Whole Prostate	Dorsolateral Lobe	Ventral Lobes	Anterior Lobes	Liver	Heart
Control	278 ± 6	375 ± 9	180 ± 18	90 ± 7	66 ± 10	24 ± 3	3833 ± 253	331 ± 15
BPH	280 ± 7	320 ± 9 **	508 ± 34 ***	220 ± 21 ***	212 ± 18 ***	76 ± 6 ***	3222 ± 265	400 ± 16
BPH + finasteride	280 ± 7	346 ± 6 *, #	382 ± 14 ***, ###	180 ± 9 ***, ##	148 ± 8 ***, ###	54 ± 3 ***, ##	3348 ± 196	375 ± 10
BPH + isoflavones	280 ± 10	317 ± 13 **	459 ± 34 ***, #	221 ± 15 ***	166 ± 19 ***, ##	75 ± 5 ***	3435 ± 225	409 ± 39
BPH + astaxanthin	280 ± 5	339 ± 11 *	443 ± 14 ***, ##	190 ± 5 ***, #	178 ± 12 ***, ##	75 ± 4 ***	3268 ± 229	387 ± 18

*, **, ***—*p* < 0.05, *p* < 0.01, *p* < 0.001 compared to control group (placebo); #, ##, ###—*p* < 0.05, *p* < 0.01, *p* < 0.001 compared to BPH group; assessed using ANOVA. BPH—benign prostatic hyperplasia.

**Table 2 plants-10-02735-t002:** Biochemical analysis of ventral lobes of the prostate of rats with BPH and treated with finasteride, kudzu isoflavones, or astaxanthin.

Group	Testosterone, NG/G Protein	Ca µmol/g Protein	Mg µmol/g Protein	Ca/Mg
Control	4.6 ± 0.9	49.0 ± 16.5	62.2 ± 11.8	0.8 ± 0.2
BPH	5.7 ± 1.7	62.6 ± 15.9	53.1 ± 15.7	1.3 ± 0.3 ***
BPH + finasteride	5.0 ± 2.8	56.9 ± 8.8	59.1 ± 8.6	1.0 ± 0.1 ##, *
BPH + isoflavones	4.7 ± 2.1	51.3 ± 2.3	63.4 ± 9.4	0.8 ± 0.1 ##
BPH + astaxanthin	5.8 ± 1.9	52.6 ± 3.9	64.8 ± 5.8	0.8 ± 0.1 ##

*, ***—*p* < 0.05, *p* < 0.001 compared to control group (placebo); ##—*p* < 0.01 compared to BPH group; assessed using ANOVA.

**Table 3 plants-10-02735-t003:** Blood count analysis of rats with BPH and treated with finasteride, kudzu isoflavones, or astaxanthin.

**Group**	**White Blood Cells, 10^9^/L**	**Lymphocytes, 10^9^/L**	**Monocytes, 10^9^/L**	**Granulocytes, 10^9^/L**	**Red Blood Cells, 10^12^/L**	**Hemoglobin, g/L**	**Hematocrit, %**	**Platelets, 10^9^/L**
Control	16.2 ± 1.9	12.6 ± 1.5	0.4 ± 0.1	3.2 ± 0.3	10.1 ± 0.2	163 ± 3	50 ± 1	954 ± 62
BPH	9.9 ± 0.9 *	5.9 ± 0.6 *	0.3 ± 0.0	4.0 ± 0.6	11.3 ± 0.2 *	182 ± 4 *	59 ± 1 *	1054 ± 74
BPH + finasteride	9.8 ± 0.4 *	5.4 ± 0.4 *	0.3 ± 0.0	4.3 ± 0.5	11.6 ± 0.2 *	192 ± 4 *	62 ± 1 *	950 ± 128
BPH + isoflavones	14.2 ± 1.1 #	9.4 ± 0.8 #	0.5 ± 0.1	4.5 ± 0.4 *	11.9 ± 0.2 *	197 ± 2 *	63 ± 1 *	951 ± 143
BPH + astaxanthin	12.2 ± 1.2	7.7 ± 1 *	0.4 ± 0	4.3 ± 0.2 *	11.5 ± 0.1 *	190 ± 3 *	61 ± 1 *	1000 ± 31

*—*p* < 0.05, compared to control group (placebo); #—*p* < 0.05, compared to BPH group.

**Table 4 plants-10-02735-t004:** Blood biochemistry analysis.

Group	Glucose (mmol/L)	Triglycerides (mmol/L)	Ca (mmol/L)	Mg (mmol/L)	Ca/Mg	SOD, Relative Activity	Testosterone, ng/mL
Control	9.2 ± 1.4	2.9 ± 0.8	2.8 ± 0.3	0.9 ± 0.1	3.2 ± 0.2	100 ± 8%	3.4 ± 1.9
BPH	7.2 ± 1.4 **	2.4 ± 1.2	3.4 ± 0.6 *	1.5 ± 0.8 *	2.7 ± 0.7	67 ± 7% *	13.5 ± 4.2 ***
BPH + finasteride	7.0 ± 0.3 **	2.2 ± 1.5	2.8 ± 0.3 #	0.9 ± 0.1 #	3.0 ± 0.2	98 ± 6% #	18.2 ± 3.0 ***, #
BPH + isoflavones	6.8 ± 0.9 **	1.7 ± 1.4	2.9 ± 0.2 #	1.0 ± 0.1 #	2.9 ± 0.1	88 ± 5%	14.2 ± 2.3 ***
BPH + astaxanthin	6.5 ± 1.2 **	1.8 ± 1.3	2.8 ± 0.3 #	1.0 ± 0.1 #	2.9 ± 0.2	111 ± 13% #	17.9 ± 3.3 ***, #

*, **, ***—*p* < 0.05, *p* < 0.01, *p* < 0.001 compared to control group (placebo); #—*p* < 0.05 compared to BPH group; assessed using ANOVA.

**Table 5 plants-10-02735-t005:** Principal compounds in the extracts used for the study.

Name	Chemical Structure	Chemical Name
Kudzu root NADES extract		
Daidzein	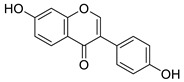	7-Hydroxy-3-(4-hydroxyphenyl)-4H-chromen-4-one
Genistein	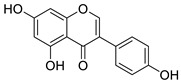	5,7-Dihydroxy-3-(4-hydroxyphenyl)-4H-chromen-4-one
Puerarin	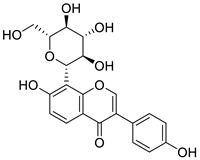	8-(β-D-Glucopyranosyl)-7-hydroxy-3-(4-hydroxyphenyl)-4H-chromen-4-one
Formononetin	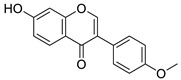	7-Hydroxy-3-(4-methoxyphenyl)-4H-chromen-4-one
Biochanin A	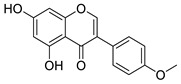	5,7-Dihydroxy-3-(4-methoxyphenyl)-4H-chromen-4-one
*Phaffia rhodozyma* extract		
Astaxanthin	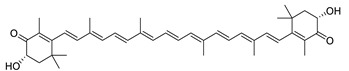	(3S,3′S)-3,3′-Dihydroxy-β,β-carotene-4,4′-dione

## Data Availability

Data is contained within the article or Appendix A.

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
