# Peer review of "Astaxantin and Isoflavones Inhibit Benign Prostatic Hyperplasia in Rats by Reducing Oxidative Stress and Normalizing Ca/Mg Balance"

_plants, 2021, doi:10.3390/plants10122735_

Round 1

Reviewer 1 Report

Dear Editor and Authors,
I have reviewed the manuscript “Astaxantin and isoflavones inhibit benign prostatic hyper-2 plasia in rats by reducing oxidative stress and normalizing 3 Ca/Mg balance” submitted to the Plants journal, part of topic “Frontiers in Phytochemicals”.  
The manuscript analyses the Astaxantin and isoflavones protective/beneficial effects on benign prostatic hyperplasia in rats.  Authors have observed that Astaxantin and isoflavones application inhibits benign prostatic hyperplasia in rats by reducing oxidative stress and normalizing Ca/Mg balance.
Please find below my comments and suggestion concerning different parts of the manuscript requiring improvement: 
Comment 1: Dear authors I would suggest the Introduction to be shortened approxımately by 30%.
Comment 2: I would suggest Authors to add some details in the explanation of The Aim (last sentence of Introduction) and make it clearly separated from the main text of introduction.
Comment 3: Authors should provide Herbarium deposition number of used plants. It will be good if available.
Comment 4: Few technical mistakes were detected please look at the uploaded manuscript file including comments.
I hope that my comments and suggestions will be useful.

Author Response

Answers to Reviewer 1

On behalf of all co-authors I would like to thank you for your valuable comments and suggestions! We did our best to address the comments accordingly and made some overall improvements throughout the manuscript.

Comment 1: Dear authors I would suggest the Introduction to be shortened approxımately by 30%.

  • We have deleted or moved to discussion some excessive parts of introduction section.

Comment 2: I would suggest Authors to add some details in the explanation of The Aim (last sentence of Introduction) and make it clearly separated from the main text of introduction.

  • The Aim was separated from introduction section and expanded.

Comment 3: Authors should provide Herbarium deposition number of used plants. It will be good if available.

  • Herbarium deposition is not widely used in Russian federation. Taking into account the importance of consistency of plant extracts from study to study, we used HPLC methods for identification and quantification of main active components for every used batch of isoflavones and astaxanthin. The explanation is added to new Study Limitations section of Materials and Methods

Comment 4: Few technical mistakes were detected please look at the uploaded manuscript file including comments.

  • Technical mistakes were corrected. Thank you for you notes.

Reviewer 2 Report

  • General comment

The manuscript entitled “Astaxantin and isoflavones inhibit benign prostatic hyperplasia in rats by reducing oxidative stress and normalizing Ca/Mg balance” evaluate the role of these two compounds in the development of prostatic hyperplasia in an animal model.

The manuscript is well written and provide an interesting field of research on plant-based compounds that are becoming constantly more important in the clinical practice. In particular, the clarity and the fluency of the text and the precision of figures and tables permit a pleasant and informative read.

Few corrections would be suggested in order to improve the overall quality of your work

  • Major corrections

Introduction

38: although BPH is associated with worse quality of life and LUTS, currently the findings regarding an increased risk of prostate cancer are still controversial. I suggest a more cautious affirmation.

51 to 60: regarding the role of compounds used, please also see https://doi.org/10.3390/nu12092648 and https://doi.org/10.3390/nu13113750. Consider those papers also for your discussion.

Results

114: although the tables are quite informative, consider to briefly report your findings also in the text

151: this section could be improved, as it seems a bit too rushed

Discussion

Overall the discussion require an improvement in terms of comparison with other similar studies in literature and report of your findings. I suggest to focus your attention on the imbalance Ca/Mg too. Finally, add the limitations of your study and potential bias that could have affected your work

  • Minor corrections

Introduction

47: consider to expand this section, as it is particularly interesting that not only oxidative stress but also chronic inflammation could have a role in an increased risk for malignancy

50: although both drugs are characterized by peculiar and wide range of adverse effects, I would like to suggest a more cautious affirmation also in this case as, excepted some particular cases, the adverse effects are manageable and limited, in relation to the advantages provided by those drugs.

73 to 77: consider to report those findings in the discussion more than in the introduction

84 to 87: as before

Methods

309: write the name of the first author at least

342: as before

Author Response

Answers to Reviewer 2

The manuscript is well written and provide an interesting field of research on plant-based compounds that are becoming constantly more important in the clinical practice. In particular, the clarity and the fluency of the text and the precision of figures and tables permit a pleasant and informative read.

Thank you very much for you attention to the manuscript and valuable suggestions, which helped improve our paper.

Major corrections

Introduction

38: although BPH is associated with worse quality of life and LUTS, currently the findings regarding an increased risk of prostate cancer are still controversial. I suggest a more cautious affirmation.

  • The sentence was rephrased for more cautious approach.

51 to 60: regarding the role of compounds used, please also see https://doi.org/10.3390/nu12092648 and https://doi.org/10.3390/nu13113750. Consider those papers also for your discussion.

  • Both reviews are addressed in the introduction section

Results

114: although the tables are quite informative, consider to briefly report your findings also in the text

  • more details are added to the appropriate manuscript section

151: this section could be improved, as it seems a bit too rushed

  • more details are added to the appropriate manuscript section

Discussion

Overall the discussion require an improvement in terms of comparison with other similar studies in literature and report of your findings. I suggest to focus your attention on the imbalance Ca/Mg too. Finally, add the limitations of your study and potential bias that could have affected your work

  • Added study limitations to the materials and methods section

  • Minor corrections

Introduction

47: consider to expand this section, as it is particularly interesting that not only oxidative stress but also chronic inflammation could have a role in an increased risk for malignancy

  • Chronic inflammation is now mentioned in the introduction section

50: although both drugs are characterized by peculiar and wide range of adverse effects, I would like to suggest a more cautious affirmation also in this case as, excepted some particular cases, the adverse effects are manageable and limited, in relation to the advantages provided by those drugs.

  • The sentence was rephrased for more cautious approach.

73 to 77: consider to report those findings in the discussion more than in the introduction

  • This part was moved to the discussion section

84 to 87: as before

  • This part was moved to the discussion section

Methods

309: write the name of the first author at least

  • The name of first author is added

 342: as before

  • The name of first author is added

Round 2

Reviewer 2 Report

No further corrections are required